

# Using Satellite Laser Ranging to measure ice mass change in Greenland and Antarctica

Jennifer A. Bonin[1], Don P. Chambers[1], Minkang Cheng[2]

[1]College of Marine Science, University of South Florida, Tampa, FL, 33701, USA
[2]Center for Space Research, University of Texas at Austin, Austin, TX, 78759, USA

*Correspondence to*: Jennifer A. Bonin (jbonin@mail.usf.edu)

**Abstract.** A least squares inversion of Satellite Laser Ranging (SLR) data over Greenland and Antarctica could extend gravimetry-based estimates of mass loss back to the early 1990s, and fill any future gap between the current Gravity

Recovery and Climate Experiment (GRACE) and the future GRACE Follow-On mission. The results of a simulation suggest that, while separating the mass change between Greenland and Antarctica is not possible at the limited spatial resolution of the SLR data, estimating the total combined mass change of the two areas is feasible. When the method is applied to real SLR and GRACE gravity series, we find significantly different estimates of inverted mass loss. There are large, unpredictable, interannual differences between the two inverted data types, making us conclude that the current 5x5

spherical harmonic SLR series cannot be used to stand in for GRACE. However, a comparison with the longer IMBIE time-series suggests that on a 20-year time-frame, the inverted SLR series' interannual excursions may average out, and the long-term mass loss estimate be reasonable.

## 1 Introduction

Since the Gravity Recovery and Climate Experiment (GRACE) was launched in 2002 (Tapley et al., 2004), it has provided

an excellent time series of mass change integrated over Greenland and Antarctica's ice sheets (Jacob et al., 2012; Luthcke et al., 2013; Schrama and Wouters, 2011; Shepherd et al., 2012; Velicogna and Wahr, 2013). However, GRACE data go back to just mid-2002, and only a few other data series exist before then to study longer-term mass change. These include satellite altimetry (Howat et al., 2008; Johannessen et al., 2005; Shepherd et al., 2012) and the 'input-output' method's combination of surface mass balance models and glacier flow speeds from interferometry (Rignot et al., 2011; Sasgen et al., 2012;

Shepherd et al., 2012). Due to the paucity of data and its limited resolution in both space and time, estimates of ice mass change before GRACE are necessarily more uncertain. High-quality Satellite Laser Ranging (SLR) tracking data [*Cheng et al.*, 2011, *Cheng et al.*, 2013] to geodetic satellites is one possible additional data set that could be exploited to compute variability in ice mass before 2002, as it exists for over a decade before GRACE.





Although SLR tracking data can be used to infer time-variable mass change [e.g., *Nerem et al.*, 2000], it can only do so over a much longer wavelength. The resolution of SLR-based gravity fields is 8000 km at the equator (based on 5x5 spherical harmonic Stokes coefficients), compared to 660 km for GRACE (based on 60x60 spherical harmonics). This difference in resolution has resulted in few ice mass studies having been completed with SLR data. For example, *Nerem and Wahr* [2011]

compared an SLR $C_{20}$ Stokes coefficient time-series with a time-series from GRACE-based estimates of Greenland and Antarctica mass loss. This led them to suggest that the two ice sheets could explain the increase in the rate of change of $C_{20}$ in the late 1990s. However, this analysis is not the same as our goals, as it used GRACE observations to explain SLR signals, rather than determining mass change directly from the SLR data. More recently, *Matsuo et al.* [2013] used a 4x4 SLR-based gravity series to demonstrate the similarities between SLR and GRACE data in a general sense. They noted

similar mass loss over the entire Arctic and showed that the center of that mass loss occurred over roughly the same spatial extent. These two examples are promising, and suggestive that SLR and GRACE may be seeing comparable signals. However, as *Matsuo et al.* acknowledged, the low spatial resolution of the SLR data makes it "not feasible to obtain definitive estimates of the total amount of the mass change… even for an area as 'large' as Greenland."

To better resolve the SLR signal and obtain a more definitive estimate than *Matsuo et al.*'s direct method, we will utilize a least squares inversion technique to localize the SLR signal over Greenland and Antarctica. This technique provides us with time-series of interannual variability, as well as decadal-scale trends and accelerations over Greenland and Antarctica. Data and methods are described in sections 2 and 3, and in the supplemental material. In section 4, we compare inversions of the SLR and GRACE data over Greenland and Antarctica during GRACE's 2003-2014 time frame, and compare their trends and

interannual signals. The implications of the results of our experiments, as well as the extension of the SLR data back to 1994, are discussed in section 5.

## 2 Data Sets

The primary data series used here are a set of maximum degree/order 60 ("60x60") monthly-averaged spherical harmonic

Stokes coefficients from GRACE (dates: 2003-2016) and a set of 5x5 monthly-averaged spherical harmonic coefficients from SLR to a series of geodetic satellites (dates: 1994-2016). A second, more limited, set of 10x10 SLR coefficients is also tested for comparison (dates: 2000-2014).

The GRACE series used here is the standard CSR Release-05 spherical harmonic version

(ftp://podaac.jpl.nasa.gov/allData/grace/L2/CSR/RL05/) (Bettadpur, 2012), with no constraints applied during processing. We apply the following standard post-processing steps: 1) $C_{20}$ is replaced with the estimate derived from SLR tracking (ftp://podaac.jpl.nasa.gov/allData/grace/docs/TN-07_C20_SLR.txt) due to GRACE's known weakness in resolving that



harmonic (Chambers, 2006), 2) a pole-tide correction is applied to harmonics $C_{21}$ and $S_{21}$ (Wahr et al., 2015), and 3) a GIA model is removed. The GIA model is composed of the W12a GIA model (Whitehouse et al., 2012) south of 62°S, and the *A et al.* [2013] model north of 52°S, using a smoothed combination of the two between 52-62°S. No smoothing or destriping [e.g., *Swenson et al.*, 2006; *Chambers and Bonin*, 2012] is applied, nor are any geocenter (degree 1) coefficients utilized. In

addition to using the full 60x60 GRACE coefficients for 2003-2014, we also truncate down to 5x5 and 10x10 subsets, to compare more directly to the SLR data.

The primary SLR series used here (Cheng, 2017; Cheng et al., 2011, 2013) is a variant of the weekly, 5x5 SLR product created at the University of Texas's Center for Space Research (CSR) and released alongside the GRACE series

(ftp://podaac.jpl.nasa.gov/allData/tellus/preview/L2/deg_5/CSR.Weekly.5x5.Gravity_Harmonics.txt). We use a version that is averaged monthly, rather than weekly, to make it more directly comparable to the monthly GRACE data. This version contains an estimate of $C_{61}/S_{61}$ (but no other degree-6 harmonics) to avoid skewing the $C_{21}$ harmonic due to a lack of sufficient degrees of freedom during the creation of the SLR gravity product (Cheng and Ries, 2017). The same GIA model is removed as with GRACE. Though the Cheng 5x5 SLR series exists from 1993 onward, prior to November 1993, only

four satellites were used in its creation (Starlette, Ajisai, and Lageos 1 and 2), whereas after that point, Stella was added as well. Because this change in satellite geometry could create possible jumps in the time-series, we have only used data from 1994 onwards. The geocenter (degree 1) SLR terms are removed, both for comparison's sake (because GRACE cannot perceive them) and because the SLR $C_{10}$ term is suspected to have an incorrect trend caused by non-uniform ground network coverage (Collilieux et al., 2009; Wu et al., 2012). The geocenter terms commonly added to GRACE (Swenson et al., 2008)

are expected to be more accurate, but they cannot be created for months when GRACE does not exist, and thus cannot be used at all before 2002. We found that using no geocenter at all brought our results closer to the results using GRACE-derived geocenter terms than using the original SLR geocenter terms did.

A pair of secondary SLR series (Sośnica et al., 2015), created at the Astronomical Institute at the University of Bern, are also

considered for comparison, though they do not extend far back in time before GRACE. Like the primary Cheng 5x5 SLR series, the two Sośnica SLR series were created from the combination of multiple satellites' SLR tracking data – mostly the five used in the Cheng 5x5 series, but also including BLITS, Beacon-C, and LARES, over the time spans they exist. Monthly solutions for 2000-2014 are available for download (ftp://ftp.unibe.ch/aiub/GRAVITY/SLR). Two versions exist: an unconstrained case to maximum degree/order 6x6, and a constrained case to 10x10. Again, the geocenter terms are not

included and the same GIA correction used in the GRACE processing is removed.

Before enacting any inversion in the spatial domain, we wish to understand how similar these three SLR series are to the GRACE series, over the limited spherical harmonics they contain. To demonstrate this, we first smooth all of time-series for each gravity coefficient with a 200-day window, thus removing signals with semi-annual periods and shorter, which are





likely to be noisy in both SLR and GRACE. We then compute the percent of the smoothed GRACE variance that is explained by each SLR series (Figure 1), via the equation:

$$PVE = \frac{1 - var(GRACE - SLR)}{var(GRACE)} \qquad (1)$$

where *var* denotes the variance of either the GRACE series or the residual once SLR is subtracted off. A percent variance

5    explained (PVE) of one means perfectly matching signals, a PVE of zero means that removing SLR does not reduce the

GRACE variance, and a negative PVE means that the residual actually has more variability than the original GRACE series

did. Ideally, we would want our PVEs to be above zero for all harmonics, and near to one for the largest and most important

harmonics.

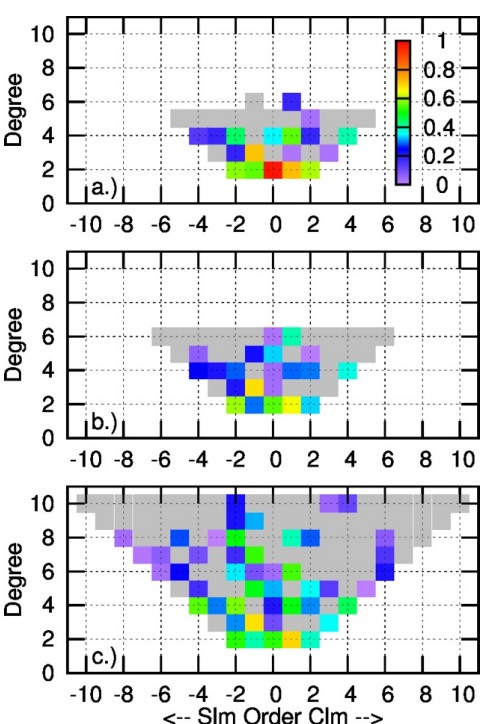

**Figure 1: Percent of GRACE variance explained by three SLR time series, after a 200-day smoother has been applied. SLR series are: (a) Cheng et al's 5x5 series, (b) Sośnica et al's 6x6 unconstrained series, and (c) Sośnica et al's 10x10 constrained series. Harmonics with negative percent variance explain are greyed out. The $C_{20}$ term in (a) is a perfect 1.0, because the GRACE $C_{20}$ has been replaced by the SLR value. S harmonics are denoted as negative orders along the x-axis, while C terms are listed as positive**

15    **ones.**

We find that around half of the GRACE signal is explained by SLR for the degree-2 harmonics, but that skill rapidly

decreases with wavelength. Above degree 4, none of the three modern SLR series explain a large percentage of the GRACE



signal. Many of the harmonics of degrees 3 and above have negative PVEs, demonstrating SLR's known low sensitivity to them. Additionally, while low-degree harmonics from truncated GRACE series are well-separated from the higher-degree coefficients, lower-degree SLR harmonics will inherently contain aliased errors from the unsolved-for higher-degrees.

The Sośnica 10x10 and Cheng 5x5 series have generally comparable PVEs at the lower degrees. While the Sośnica 6x6 data is similar to the Sośnica 10x10 data at degrees 2-3, it explains significantly less of the GRACE variance for degrees 4-6. For that reason, we focus on the other two series in this paper. The Cheng 5x5 series is particularly useful in this study because of its much longer record, but the independent nature of the Sośnica 10x10 makes it valuable for comparison.

**3 Methods: Global Inversion**

To localize the mass signal from the low-resolution GRACE and SLR series into areas near Greenland and Antarctica, we use a modified version of the inversion technique described in *Bonin and Chambers* [2013]. In that paper, a series of regions are defined ahead of time, and a least squares approach constrained by process noise is used to estimate the amount of mass change arising in each region. We attempted to use the same approach here, but quickly found that what can be done with

60x60 data sets cannot be accomplished with lower-resolution 5x5 data (see supplemental information).

Instead, we use a correlation-based approach to constrain the least squares inversion. We first separate the world into three main areas: Antarctica, the ice-covered area near and including Greenland, and everything else. We divide each large area into multiple sub-regions, then tie those sub-regions loosely together with spatial and temporal constraints. This allows

different sub-regions, such as eastern vs. western Antarctica, to vary at different times, while still keeping the number of observations significantly greater than the number of independent parameters solved for, thus giving a stable solution. The constraints are based on the JPL mascon GRACE data (Watkins et al., 2015) from 2003 to 2014, after GIA has been removed. We compute cross-correlations between sub-regions within each area from the mascon data, and use those to constrain the sub-regions to vary in expected spatial patterns. We also use lag-1 auto-correlations of each sub-region to force

each month's solution towards the neighboring months'. The derivation of the constrained inversion process is given in the supplemental information.

We first tested the process on a completely simulated data set, similar to the one used in *Bonin and Chambers* [2013]. The details of the simulated data are given in the supplemental material. The results suggest using a correlation-constrained least

squares inversion allows for accurate estimates of the Greenland and Antarctic mass change when using 60x60 or even 10x10 simulated data. However, a 5x5 resolution proves insufficient to invert the sub-annual signals correctly (Figure 2a and b). We believe that this inaccuracy comes about because both Greenland and Antarctica are polar areas, and thus



heavily dependent upon the same very low-degree spherical harmonics. Without higher-degree harmonics to clarify the situation, the mathematics cannot always determine which region to place which signal in.

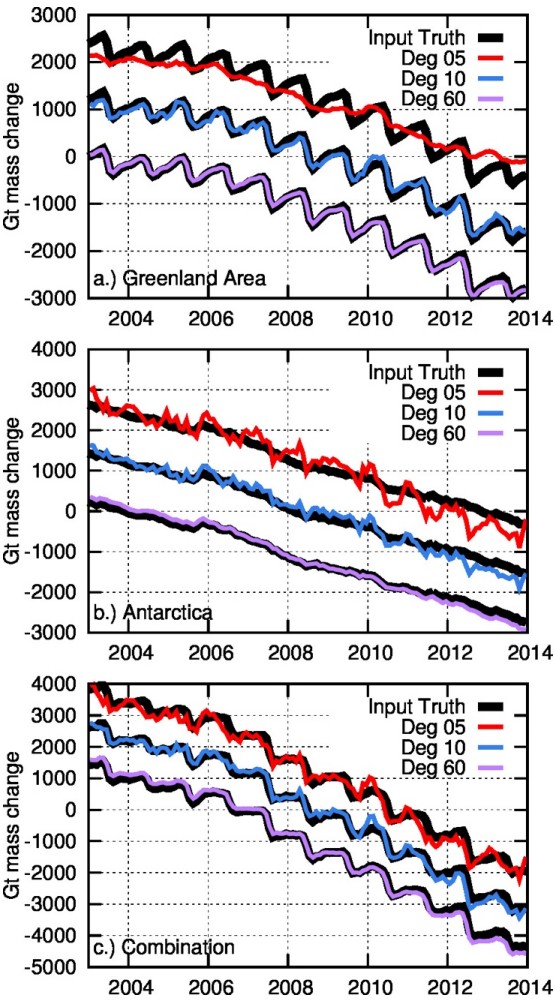

5   **Figure 2: Simulated inversion results by maximum degree/order, relative to input 'truth' signal. Regions considered: (a) Greenland and surrounding islands; (b) Antarctica; (c) the sum of Greenland and Antarctica. Each inversion was run using correlation-based constraints. Time-series are offset for clarity.**

We can eliminate this problem by summing the time series of the two areas and looking at the total mass loss over Antarctica

10   and the near-Greenland area combined (Figure 2c). Using SLR-like 5x5 harmonics for the simulation results in a negligible





simulated trend error (7 ± 18 Gt/yr). The 60x60 simulated inversion produces a small trend error of 36 ± 8 Gt/yr (6.5% the simulated 'truth' trend). After removing these trends, the remaining RMS error of the correlation-constrained simulation inversion is 202 ± 10 Gt for 5x5 data, 131 ± 10 Gt for 10x10 data, and just 37 ± 5 Gt for 60x60 data, which demonstrates that higher-resolution series are much better able to track the month-to-month variability within the data. (All errors given are

95% confidence levels, based on a Monte Carlo simulation of random noise with a known red spectrum, after fitting for a bias, trend, annual, and semi-annual signals. The Monte Carlo simulation's values are generated using the same RMS and lag-1 autocorrelation as the inverted data.)

### 4 Analysis: Comparison with GRACE

Based on the results of the simulation, we applied the least squares inversion technique with correlation-based constraints to the real SLR and GRACE data and summed over all of Antarctica and the near-Greenland area. The resulting mass change time-series are shown in Figure 3. For a comparison 'truth' signal, we use a combination of two higher-resolution inversions of the 60x60 GRACE data, which inverts over only Antarctica and Greenland individually, and places each local signal into more, smaller regions. This technique more accurately estimates the mass trends and higher-resolution signals than the

larger-region correlated technique can, since its regions and parameters are tuned for the full 60x60 data rather than 5x5 data (see supplemental information). This allows for a more realistic estimate of the SLR errors. Also, since part of our goal is to match up the SLR time-series with a high-quality GRACE one, learning the mismatch between them is important all on its own.

We first consider the errors implicit in reducing the locally-defined, high-resolution GRACE inverted series (black line in Figure 3a) to a 5x5 truncated series (orange line). We find an error of 31.7 Gt/yr in trend (7.0 ± 2.5% of the high-resolution GRACE trend), such that between 2003-2014, the 5x5 GRACE inversion estimates 380 Gt greater total polar mass loss. Over that same time, the remaining RMS difference between the 5x5 and high-resolution GRACE inverted signals after the trends are removed is 220 Gt (63.7%). These numbers are fairly comparable to our 5x5 simulation-based errors of 1.3 ±

1.6% for trend and 75.1% for RMS. We should thus expect to see errors on this level from any SLR series, simply due to the signal truncation effect.





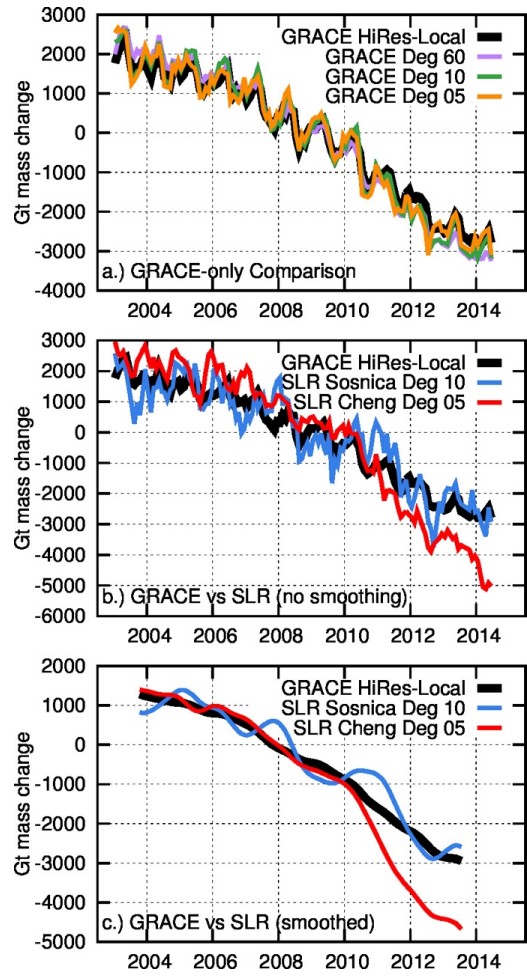

**Figure 3:** Comparisons of inverted GRACE and SLR mass signals, over Greenland and Antarctica combined. **(a) GRACE-only comparison, for different maximum degree/orders, relative to the high-resolution, local GRACE inversion. (b) SLR comparison. (c) Low-pass SLR comparison, after applying a 400-day (13 month) smoother.**

Figure 3b shows the inversion of the SLR series compared to GRACE, over only those months where both SLR and GRACE data exist. The trend differences between GRACE and the Cheng 5x5 SLR series are particularly startling (40.9 ± 11.1% error), especially considering that the Sośnica 10x10 time-series has a trend error of similar size to what simple truncation to 5x5 harmonics causes (7.3%). However, when the trend is removed, large and different RMS errors (145-167%) remain in

10    both. We smoothed both the GRACE and SLR time-series with a Gaussian smoother that cuts off periods shorter than 13



months (Figure 3c; final column of Table 1), to remove month-to-month jitter and get a better view of what is causing the differences.

In terms of trends, from 2003-2010, the Cheng 5x5 trend errors are statistically indistinguishable from zero. Then, from
2010-2014, the Cheng SLR and GRACE trends diverge suddenly and significantly (106.1 ± 28.6% trend difference). Collectively, this results in the 40.9% error from 2003-2014. The Sośnica 10x10 inversion shows no such sudden change in behavior. This divergence in the Cheng SLR data seems so sudden that we initially believed it might have been caused by the pole-tide error discussed by *Wahr et al.* [2015]. Their correction is a two-piece affair, treating the $C_{21}$ and $S_{21}$ harmonics differently before and after 2010, and its impact is largely linear. However, after applying the correction to our GRACE
data, we realized that no pole-tide correction is large enough to explain the differences we see between GRACE and the Cheng SLR series. As *Wahr et al.* noted, the impact of their correction is on the order of 0.5 cm/yr equivalent water thickness in trend throughout the world. Trends in Greenland and Antarctica are two or three orders of magnitude greater than that.

| Series to Difference, Relative to GRACE High-Res Series | Trend Error (Gt/yr) | Trend Error (%) | Residual RMS Error | Residual RMS Error (Smoothed) |
|---|---|---|---|---|
| GRACE 5x5 | -31.7 ± 11.5 | 7.0 ± 2.5 | 63.7% | 46.1% |
| GRACE 10x10 | -45.3 ± 11.3 | 10.0 ± 2.5 | 52.6% | 39.6% |
| **SLR Cheng 5x5** | **-184.8 ± 50.5** | **40.9 ± 11.1** | **145.2%** | **156.1%** |
| SLR Sośnica 6x6 | -182.2 ± 54.5 | 40.4 ± 12.0 | 188.9% | 165.1% |
| **SLR Sośnica 10x10** | **33.1 ± 31.3** | **-7.3 ± 6.9** | **167.3%** | **158.0%** |

**Table 1: Differences relative to GRACE 60x60 high-resolution, local inversion, over the combined Greenland/Antarctica region during 2003-2014. Residual RMS errors are those after the trend has been removed, relative to the GRACE 60x60 detrended RMS. The final column is the residual RMS error after a 13-month Gaussian filter has been applied to all series. Errors given are purely statistical 95% confidence levels after fitting for a bias, trend, annual, and semi-annual signals, based on a Monte Carlo simulation of random red noise with the given RMS and lag-1 autocorrelations. They do not include the intrinsic errors of the**
**satellites themselves, or the effects of the inversion method. Errors are computed on series including only those months estimated by GRACE.**

So instead of being a true error in trend, the large interannual differences between GRACE and the Cheng 5x5 SLR series are probably indicative of a systematic interannual-scale error in the SLR inversion. Continuing the series past 2014 (Figure
4) encourages us in this belief, since the SLR series measures effectively zero trend in mass change for 2014-2016, bringing it back towards the GRACE series. The Sośnica 10x10 series also differs significantly from GRACE on the interannual scale, despite the good agreement in trend. Its pattern of difference is more sinusoidal, with 2- to 3-year periods, on top of a small but more-or-less constant trend difference. On an even shorter scale, the Cheng and Sośnica SLR series both resolves large annual-scale and shorter fluctuations that GRACE does not see. Since the SLR series do not see the same changes in




either annual or multi-year signals as either each other or GRACE, we presume that the differences are most likely errors in SLR, though it is possible that GRACE contains unsuspected large interannual errors as well.

## 5 Results: 1994-2017 Time-Series

It is disappointing but not a tremendous surprise that the SLR series cannot fully resolve the varying nature of the polar mass signal. GRACE is a rather high-resolution data set, while as Figure 1 demonstrates, only the lowest-degree part of the SLR estimates are likely to be highly accurate. Our simulation showed that we are already pushing at the bounds of our spatial resolution to try localizing 5x5 data into even a single Greenland and Antarctic region, so one presumes that combining that difficulty with incorrect higher-degree values in SLR results in the large interannual errors that we see. Certainly, those

errors mean that a 5x5 SLR field cannot be used to fill in gaps in the GRACE/GRACE Follow-On record.

However, in a longer-term sense and bearing in mind the limitations of the data, SLR does a fair job of estimating ice mass change. The Sośnica 10x10 series is not available much before GRACE or after 2014, but we can compute the Cheng 5x5 SLR inversion back to 1994 and through to the beginning of 2017 (Figure 4). The most recent years of data show that the

sharp divergence beginning in 2010 is recovering by 2017. (The lack of other satellite or in-situ evidence for an increased mass loss from 2010-2014, and a stable mass state since then, makes us certain that SLR is less accurate than GRACE over this time-span.) If this recovery continues, it will represent not a trend error, but an interannual error with a divergent period of around five years. Given that suggestive evidence, it is possible that the Cheng SLR series might be broadly accurate on the 1994-2017 time-scale, even though any individual year's estimate could be fairly far off.


The Cheng 5x5 SLR series' constant twenty-three-year trend is -451 ± 28 Gt/yr. However, a single line is an extremely poor approximation for this longer, sharply curving data set. If we instead assume that the ice sheets are in a long-term stable state at the beginning of 1994, then we can determine a constantly accelerating curve at an optimal point along the 1994-2017 SLR data (orange line in Figure 4). The best two-piece fit to the data involves a constant (zero mass change) part until

December of 1996 (± 5 months) followed by a constant acceleration of -25.8 ± 1.1 Gt/yr$^2$ thereafter. As Figure 4 shows, even this model exaggerates the amount of mass that SLR sees lost after 2016 – an effect which would not occur if the Cheng SLR series did not diverge from GRACE beginning in 2010.



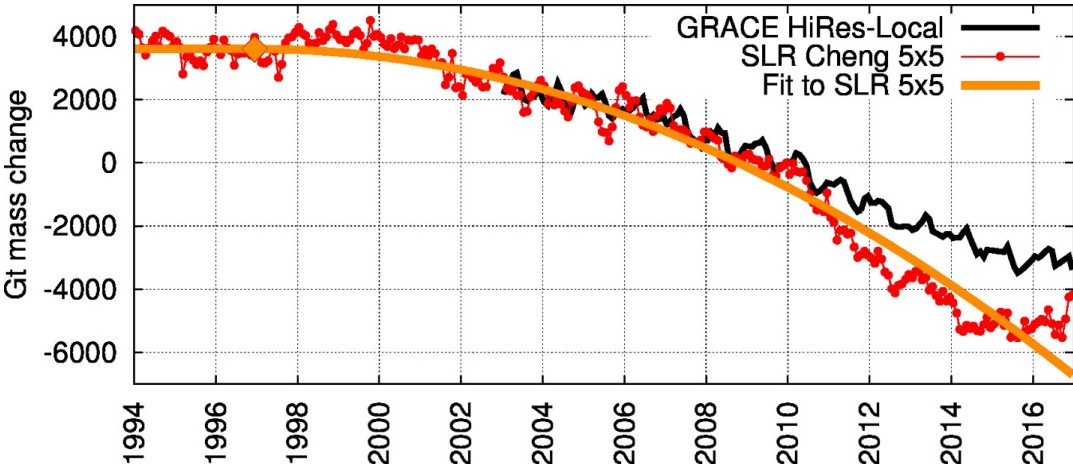

**Figure 4: Mass loss over Greenland and Antarctica combined, carried back to 1994, from the Cheng 5x5 SLR inversion. Monthly results are shown as red dots, with the best-fit accelerating curve sketched in orange. The orange diamond represents the point at which acceleration begins. The high-resolution, local GRACE inversion is shown (black) beginning in 2003, for comparison.**

The obvious question we need to answer is how often SLR takes such multi-year excursions, and whether it really does get back on track afterwards. One way to get a feel for the pre-GRACE accuracy of the SLR inversion is via a comparison with an additional data set. The Ice-sheet Mass Balance Inter-comparison Exercise (IMBIE) for Greenland and Antarctica (http://imbie.org/data-downloads) (Shepherd et al., 2012) is a time-series of mass change created from a combination of different techniques and data sources. It is based on the model-based input-output method and radar altimetry before 2003

and on the input-output method, laser and radar altimetry, and GRACE after 2003. It does not exist over the islands near Greenland which we included in our estimate, principally including Iceland, Svalbard, Ellesmere Island, and Baffin Island. To make a fair comparison, we mask out these neighboring islands from our final gridded solution, so as to compare across the same area, then compute the summed mass change over Antarctica and Greenland. We also smooth both GRACE and SLR with a 13-month Gaussian smoother to duplicate what was done with IMBIE. One significant difference remaining is

that IMBIE naturally includes the impact of the geocenter terms, while we have excluded those from our SLR estimate because of their large expected errors.



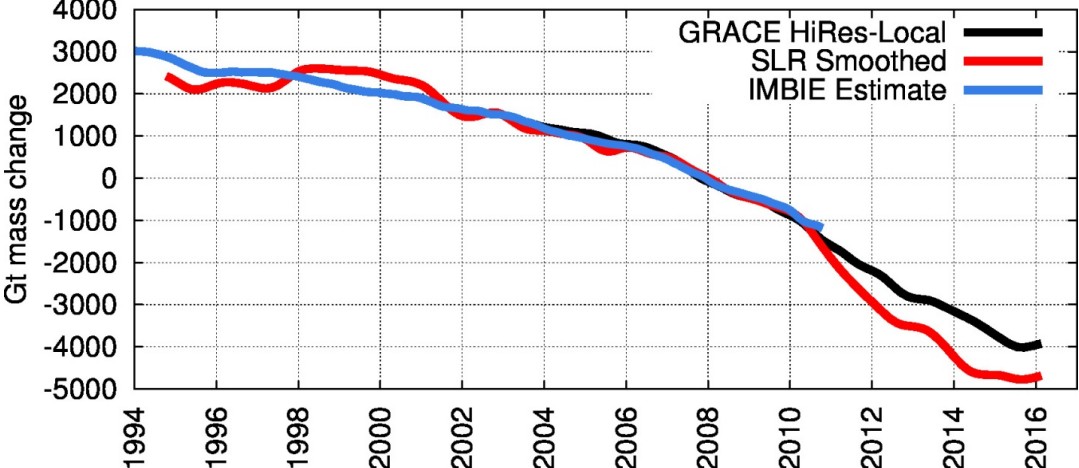

**Figure 5:** The high-resolution localized GRACE (black), Cheng 5x5 SLR (red), and IMBIE (blue) estimates of Greenland and Antarctica's mass change. A 13-month smoother has been applied to the GRACE and SLR results, and they are scaled to include only the areas of Antarctica and Greenland, not the islands surrounding Greenland, to duplicate the IMBIE approach.

As Figure 5 demonstrates, IMBIE's mass change estimate aligns neatly with GRACE during its six-year overlapping time-span, but also approximates a similar long-term signal to SLR before GRACE. During the overlapping fifteen-year period (1994-2009), the Cheng 5x5 SLR inversion estimates an average mass loss rate of -197 ± 40 Gt/yr, while IMBIE sees a statistically identical trend of -220 ± 42 Gt/yr. (The uncertainty here is based on the variance of the smoothed residuals

about the fit, but also accounts for temporal correlation due to the 13-month smoothing already applied to the IMBIE data. This reduces degrees of freedom from 186 to 14, so inflates the error from the least squares fit by sqrt(186/14).) Assuming IMBIE is correct, the SLR inversion sees multi-year errors before 2002, as it does from 2010-2017. However, over the long-term, these errors have averaged out before, as they seem to be in the process of doing now.

**6 Conclusion**

Because of the large uncertainty on interannual periods, we do not believe this inverted SLR data series should be used to estimate mass loss over Greenland and Antarctica on its own. Certainly, we cannot use it to fill short-term gaps in the GRACE record, or between GRACE and the future GRACE Follow-On mission. Nonetheless, over longer time spans (~20 years), the inverted Cheng 5x5 SLR series appears to measure real mass change signal, similar to the more extensive IMBIE

estimates, and thus ought to be considered in combination with other data sources in the future. As an attempt to make SLR more useful for this effort, our future work will include the creation of a new SLR series, created in the same manner as the

Cheng 5x5 series, but including a year of data in each estimate, rather than a month. The hope is that by sacrificing the sub-annual signal, we can gain better accuracy at inter-annual periods, thus reducing the variability which stymies us here and creating a more useful pre-GRACE estimate of total mass change over Greenland and Antarctica.

**7 Acknowledgments**

We would like to deeply thank John Ries at the University of Texas's Center for Space Research for his kind assistance in the preparation of this paper. Thank you so much for sharing your generous SLR background knowledge and advice with us.

This research was conducted under a New (Early Career) Investigator Program in Earth Science NASA grant
(NNX14AI45G). We are most appreciative of NASA's funding and support.

**8 Data Availability**

The monthly Cheng 5x5 SLR data is available as part of the supplemental information, online at doi:10.5281/zenodo.831745. All other data series are publically available at the websites listed in the text. The numerical
inversion results or mapped regional definitions are available from the authors upon request.

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
