# Peer review of "Using Satellite Laser Ranging to measure ice mass change in Greenland and Antarctica"

_The Cryosphere, 2017_

## Short Comment (SC1) · 25 Jul 2017

The paper is a helpful exploration of the potential, and limitations, of using Satellite Laser Ranging to extend GRACE gravity field time series backward in time to the early 1990s. It's a good basis for future work to explore the propagation of systematic error and try and improve on these estimates in the future. I wasn't clear of a few things which I personally would appreciate if the authors could clarify.

The $C_{2,0}$ term of GRACE is replaced by the SLR $C_{2,0}$ (from the Cheng 5x5 solution?). This results 100% of the variance explained in Fig 1a (although I note percents are not shown since the scale is 0->1) but lower variance explained when comparing to the

[Figure]

Sosnica solution. Is that correct and if so, is it a fair comparison. PS beware rainbow colour scales (https://www.climate-lab-book.ac.uk/2016/04/)

Given the duplication of the C2,0 term, should not it be excluded from the comparison to GRACE?

it would be good to see in the supplement degree,order specific time series comparisons for GRACE and SLR to see where the differences occur.

I wasn't sure if autocorrelation was really treated correctly - the authors assume it is diminished by 13-month averages and reduce the degrees-of-freedom appropriately but I think the assumption the series is white noise after this averaging (ie, uncorrelated). Exploration of the noise model by examining the spectra and fit of various noise models could be worth considering although I see an argument here that an exact specification of uncertainty is not the key message but the bias magnitudes. If interested in this see Williams et al EPSL 2014 for example - there's some nice tools available to test different noise models; see HECTOR (Bos et al) or est_noise (Langbein et al) for example. By the way, I don't think 13-month averages really reduce all signal with periods less than 13 months to zero.

anyhow, I thought it an interesting paper and hopefully these remarks contribute to the authors' thinking in a constructive way

Matt

---

## Referee Comment (RC1) · Anonymous Referee #1 · 25 Aug 2017

General Remarks:

With interest I've read the manuscript by Bonin et al. The manuscript describes 2 inversion methods of Satellite Laser Ranging (SLR) data, in order to solve for mass changes in Antarctica and Greenland. The inversion method's ability to recover the complete signal is tested with a simulation (with 'perfect' data based on models). Results with real data are compared to GRACE over the GRACE time period. For comparison two SLR solutions are compared, each with a different spherical harmonic truncation degree (5 versus 10).

I found the paper easy to read, and appreciate the transparent assessment of the

capabilities of SLR. Together with the supplement, I can imagine that the paper is informative for readers of the Cryosphere, especially those who have an affinity with geodesy. It has to be said that a related piece of work has recently appeared (Talpe et al. 2017), but from my point of view the differences w.r.t. this paper are large enough to justify the publication of this work. Nevertheless, there are a few remarks, which I think need to be addressed before accepting the manuscript.

* Replacement of C20 by SLR derived estimates This issue has already been mentioned by Matt King in his short comment. So this is to reconfirm that this issue also stroke me as somewhat tricky. By replacing C20 by an SLR-estimate a dependency is introduced which may be favorable for the CSR-SLR solution in the comparison. To clear this up, maybe the authors could show how much C20 contributes to the estimated time series.

* Use of diagonal SLR and GRACE error-covariances , and thus neglecting off-diagonal error-covariance. I think this is the most serious issue I can find in the paper. Since I don't know whether this is going to have a large impact on the results I'm recommending a major revision to allow the authors to clarify this. I suspect that in particular SLR may have significant off-diagonal components in its error-covariances. The SLR network is very sparse and may not be optimal for the retrieval for ice mass change signals at higher latitudes. To account for this, one would in principle need to propagate the full SLR error-covariance on the 1x1 degree grid used as observations. The associated error-covariance matrix of the gridpoints will consequently be quite unstable (e.g. from 36 SLR 'observations' one produces 360x180 observations, without adding more information), which potentially could break down the inversion scheme as it is implemented now. In the current setup, the authors ignored error-covariances and by choosing an equidistant 1x1 grid also artificially increased the density of observations at higher latitudes inversely proportional to cosine(lat). In a broad sense, ignoring off-diagonal contributions and artificial increase of observations can be interpreted as a regularization, which the authors should justify. I therefore, propose that the authors either justify their choices for the 1x1 grid in combination with a diagonal error-covariance or better: that the authors replace matrix H (see eq S1) by an operator which directly maps Stokes coefficients to the unknown vector a. When full error-covariances are available these can then also be implemented with hopefully relatively little effort.

* Neglecting degree 1 contributions I understand the decision of the authors to not account for the degree 1 signal, based on remaining errors in the SLR data. However, the potential influence of degree 1 neglection may be too large to ignore. As an alternative, maybe the authors can treat the degree 1 signal as noise and assess its influence on the results by producing an ensemble of realistic variations and propagating this through the inversion?

Minor details *The supplement has a *.zip ending but actually is in *tgz format

* abstract: maybe add some numbers in the abstract to quantify things a bit more

* Does the average TC reader knows what is meant by 5x5, 10x10?

* eq 1 shouldn't the '1-' be outside of the fractionr?

* "and thus heavily dependent on the same very low degree spherical harmonics" Maybe quantify this with formal error correlations?

* " indicative of a systematic interannual-scale error in the SLR inversion" What is meant by this? Maybe add a reference, which illustrates the problem at hand?

* "451 +1 Gt/yr" I assume this is for Antarctica and Greenland? Maybe explicitly mention this again

References: Talpe, M.J., Nerem, R.S., Forootan, E., Schmidt, M., Lemoine, F.G., Enderlin, E.M., Landerer, F.W., 2017. Ice mass change in Greenland and Antarctica between 1993 and 2013 from satellite gravity measurements. J Geod 1–16. doi:10.1007/s00190-017-1025-y

---

## Referee Comment (RC2) · K. HEKI (Referee) · 26 Sep 2017

Review of "Using Satellite Laser Ranging to measure ice mass change in Greenland and Antarctica" submitted to The Cryosphere by J. A. Bonin et al.

As a whole

Considering that the GRACE satellite system can go back only to 2002 in time, the authors used data sets of SLR including not only $C_{20}$ but coefficients up to higher degree/orders. They validate their method by comparing the data during periods with overlap with GRACE, and explore mass changes in Greenland and Antarctica in 1990s. They found that the mass changes over the Greenland and Antarctic ice sheets cannot be separated with the 5 x 5 model, but their sum can be discussed. The authors present results in Figure 4, which looks somewhat similar to the Greenland ice loss trend by 4 x 4 model in Matsuo et al. (2013). The result shows insignificant mass loss during 1990s and its acceleration toward the present time. This is an interesting study, and the manuscript is well written. In addition to minor comments at the end of this review, I would like to ask two major questions which need to be discussed (need not be solved) in the revised version.

No. 1 Separation of Greenland and Antarctica using external information

The limited spatial resolution of the SLR 5 x 5 model could not separate ice losses from the two ice sheets. Nevertheless, I think there are external clues to answer the question, how much coming from Greenland and how much from Antarctica. Matsuo et al. (2003) used the quadratic component in the vertical position time series of GNSS stations in Greenland to validate their results. Because of uncertainties in GIA models, it is not straightforward to discuss linear uplift/subsidence rates of the Antarctic GNSS stations. However, because GIA rates do not change in a short time-scale, quadratic (or higher degree) components in vertical position would entirely reflect the elastic response of the lithosphere to the present-day ice melting. Several GNSS station in Antarctica have been operational since 1990s, and the authors at least discuss if the signature of the accelerated ice mass loss ever exists in Antarctica.

No 2. Reality of the departure of SLR data from GRACE

Below I compare Figure 4 (left) and a figure drawn by the reviewer using the CSR Level-2 RL05 spherical harmonics data with standard filters (right). It shows the gravity time series at a certain point in southern Greenland (65N 40W), and indicate anomalous changes after 2012, a short-term accelerated mass loss in 2012 and a longer-term stationary behavior until present (reflecting increased precipitation there). I see some

similarity between the 5x5 SLR data (rather than GRACE HiRes-Local) and the mass changes in southern Greenland. Is it conceivable that mass signals in southern Greenland leaked into the SLR 5x5 solution?

[Figure]

Minor comments

Page 9 line 4: "trend errors are statistically indistinguishable from zero." sounds strange (trends could be indistinguishable from zero but errors should not be indistinguishable from zero).

Page 11 line 9: Please explain the "input-output method"?

Page 12 line 13: "before" what (words missing)?

Page 14 line 19: Nerem and Wahr (2011) missing in the reference list

---

## Author Response (AR1)

Response to Reviewer #1 (Matt King):

Thank you so much for your helpful review. Sorry it's taken me so long to get back to you on it. Let me see if I can respond to each of your main points.

1.) "Given the duplication of the C2,0 term, should not it be excluded from the comparison to GRACE?"

I think this is a really good point, since the influence of C20 is so large at the poles. You're right that using very similar C20 terms for GRACE and the Cheng SLR series might bias them toward each other for reasons that have nothing to do with GRACE itself. However, because the C20 terms are such a big part of the final signal, I didn't really want to produce this paper by totally excluding it. Instead, to answer your question, I decided to test what the impact of removing it was, to see if it was reducing the divergence I see between GRACE and the Cheng SLR series.

So I recreated each of the three main series (GRACE, Cheng 5x5 SLR and Sosnica 10x10 SLR) and totally omitted the C20 terms, then inverted each and took a look at the time series. If the C20 term was causing falsely alignment with GRACE, I would see a larger divergence between GRACE and the Cheng series, in which case, my paper would require revising.

[Figure]

**Figure 1: Left-hand image is the inversion with C20 included. Right-hand image is the inversion with C20 totally removed.**

However, I see no notable changes in terms of divergence. There are three main effects of removing the C20 terms. First, the overall trend of all three of the series dropped like a rock. (No surprise, given the geometry of the situation.) Second, the month-to-month jitter in all three of the series changed. Third, most oddly, removing the C20 term from the Cheng series produced a large, visible annual signal before about 2007. The other series (including GRACE, using a similar C20) didn't show this impact. So that's bizarre. I assume that the C20 term in the Cheng series is coupled with

some other term, to produce this (which wouldn't especially surprise its creators, since they're aware of the general coupling between harmonics caused by a barely solvable problem).

In any case, there was not any significant change in the interannual signal divergence. So in practice, the replacement of the GRACE C20 should bias GRACE towards the Cheng SLR series doesn't seem to have any major effect on the part of the spectrum that I'm worried about. That's a relief.

I have created the following commentary for the final version of the paper, briefly discussing this:

We did consider the impact of replacing the GRACE $C_{20}$ term with that from a series related to the Cheng 5x5 SLR data. To test whether this unfairly biased the Cheng 5x5 SLR results toward GRACE, we removed the $C_{20}$ terms completely from all of the GRACE and SLR series, then inverted each of them again. Removing the impact of the equatorial bulge did greatly reduce the trend of each Greenland+Antarctica inverted series, but it did not significantly impact the interannual differences between GRACE and any SLR series. We thus conclude that the replacement of GRACE's $C_{20}$ values is not a large contributing factor to these results.

Again, I want to thank you for this idea, since it was certainly a troublesome possibility.

2.) "It would be good to see in the supplement (degree, order)-specific time series comparisons for GRACE and SLR to see where the differences occur."

I have created this visual comparison (up to deg/ord 5) and will add it in the appendix for the final paper. For your immediate edification, here they are:

[Figure]

[Figure]

**Figure 2:  GRACE is in black, Cheng's SLR is in red, Sosnica's SLR is in blue.**

3.)  You are correct that Figure 1 in the main document (the percent variance explained) was given in terms of the proportion of the signal from 0 to 1, not as a real percentage.  That's been changed, so the values in the figure go from 0 to 100%.  I'll update for the final version of the paper.  Thanks.

4.)  "I wasn't sure if autocorrelation was really treated correctly - the authors assume it is diminished by 13-month averages and reduce the degrees-of-freedom appropriately but I think the assumption the series is white noise after this averaging (ie, uncorrelated). Exploration of the noise model by examining the spectra and fit of various noise models could be worth considering although I see an argument here that an exact specification of uncertainty is not the key message but the bias magnitudes…"

The error statistics given for our own solutions (ie: table 1) contain the assumption that the residual solution (after the mean, trend, annual, and semiannual terms are fit and removed) still contains a correlating signal. We assume an AR-1 method and estimate the errors based on that assumption. Based on the paper you recommended, this seems to be a reasonable assumption for Antarctica, and presumably Greenland as well.

I think, though, that maybe the statistic you were really worrying about was the comparison with the IMBIE data? Referring to that, we wrote: "The uncertainty here is based on the variance of the smoothed residuals about the fit, but also accounts for temporal correlation due to the 13-month smoothing already applied to the IMBIE data. This reduces degrees of freedom from 186 to 14, so inflates the error from the least squares fit by sqrt(186/14)."

[Figure]

**Figure 3: IMBIE inversion over GL+Ant, after removing a seven-parameter fit.**

The sqrt(186/14) assumption described here only refers to the treatment of the IMBIE data, and is based on their claims of a 13-month temporal smoothing. A quick look at the IMBIE data after removing the acceleration, trend, annual, etc (above) shows a 3-4-year quasi-periodicity remaining, so I definitely agree that the signal left isn't actually white noise. To test whether the degrees-of-freedom reduction we used (based on a 13-month averaging) is "close enough", I computed the autocorrelation of the monthly IMBIE data (black line below). At a 13-month lag, the autocorrelation is 0.2. It actually crosses zero at about 16 months. I also checked to be sure that decoupling the "monthly" data points from the neighboring ones by using only every 12$^{th}$ point (colored lines) doesn't impact the autocorrelation significantly – and it doesn't.

[Figure]

IMBIE autocorrelation test

[Figure]

**Figure 4:** **The colored lines are the autocorrelation using only every 12th point, which should not have been made dependent on each other due to the temporal smoothing. The black line is the autocorrelation using all the points.**

So, technically, we should probably use a ratio of sqrt(186/11.6), leading to a weighting of the errors of 4.0 rather than 3.6. But since increasing the IMBIE errors won't impact the overall results of the paper, I doubt the detail is worth explaining the added complexity to readers (as you noted). If you feel strongly about this, though, we can change it.

10 (Overall, by the way, I agree with the paper you recommended: we too often assume that everything other than the mean, trend, annual, semiannual, and tidal aliases is "noise". Some sort of assumption for a low-frequency correlation seems more logical to me. Thanks for the link to the paper. I found it pleasantly clear to read, for a stats paper. Nice work.)

15 Thank you again for your excellent review. We appreciate the help!

    -- Jennifer Bonin
* * *
*It has to be said that a related piece of work has recently appeared (Talpe et al. 2017)…*

5   ---

Yes.  I also recently read this paper (and talked to the author), and put a brief note in the methods section of my appendix about the differences of method and similarities of general results that we found.

10   ---
*Replacement of C20 by SLR derived estimates This issue has already been mentioned
by Matt King in his short comment. So this is to reconfirm that this issue also
stroke me as somewhat tricky. By replacing C20 by an SLR-estimate a dependency
is introduced which may be favorable for the CSR-SLR solution in the comparison. To*
15   *clear this up, maybe the authors could show how much C20 contributes to the estimated
time series.*
* * *
As I also said to Matt, I think this is a good point, since the influence of C20 is so large at the poles.  You're both right
20   that using very similar C20 terms for GRACE and the Cheng SLR series might bias them toward each other for reasons that have nothing to do with GRACE itself.  However, because the C20 terms are such a big part of the final signal, I didn't really want to produce this paper by totally excluding it.  Instead, to answer your question, I decided to test what the impact of removing it was, to see if it was reducing the divergence I see between GRACE and the Cheng SLR series.

25   So I recreated each of the three main series (GRACE, Cheng 5x5 SLR and Sosnica 10x10 SLR) and totally omitted the C20 terms, then inverted each and took a look at the time series.  If the C20 term was causing falsely alignment with GRACE, I would see a larger divergence between GRACE and the Cheng series, in which case, my paper would require revising.

[Figure]

**Figure 5: Left-hand image is the inversion with C20 included.  Right-hand image is the inversion with C20 totally removed.**

However, I see no notable changes in terms of divergence. There are three main effects of removing the C20 terms. First, the overall trend of all three of the series dropped like a rock. (No surprise, given the geometry of the situation.) Second, the month-to-month jitter in all three of the series changed. Third, most oddly, removing the C20 term from the Cheng series produced a large, visible annual signal before about 2007. The other series (including GRACE, using a similar C20) didn't show this impact. So that's bizarre. I assume that the C20 term in the Cheng series is coupled with some other term to produce this (which wouldn't especially surprise its creators, since they're aware of the general coupling between harmonics caused by a barely solvable problem).

In any case, there was not any significant change in the interannual signal divergence. So in practice, the replacement of the GRACE C20 should bias GRACE towards the Cheng SLR series doesn't seem to have any major effect on the part of the spectrum that I'm worried about. That's a relief.

I have created the following commentary for the final version of the paper, briefly discussing this:

We did consider the impact of replacing the GRACE $C_{20}$ term with that from a series related to the Cheng 5x5 SLR data. To test whether this unfairly biased the Cheng 5x5 SLR results toward GRACE, we removed the $C_{20}$ terms completely from all of the GRACE and SLR series, then inverted each of them again. Removing the impact of the equatorial bulge did greatly reduce the trend of each Greenland+Antarctica inverted series, but it did not significantly impact the interannual differences between GRACE and any SLR series. We thus conclude that the replacement of GRACE's $C_{20}$ values is not a large contributing factor to these results.
* * *
*Neglecting degree 1 contributions I understand the decision of the authors to not account for the degree 1 signal, based on remaining errors in the SLR data. However, the potential influence of degree 1 neglection may be too large to ignore. As an alternative, maybe the authors can treat the degree 1 signal as noise and assess its influence on the results by producing an ensemble of realistic variations and propagating this through the inversion?*
* * *
I previously ran a comparison of the Cheng SLR geocenter terms compared to those computed over the GRACE time-period with the technique of Swenson et al. I was surprised to find that the difference between the two geocenter estimates (in terms of their impact on the inverted timeseries) was about the same size as the difference between not using any geocenter and using the Swenson version. According to my coauthor Minkang Cheng and his colleague John Ries, much of this difference is likely to be an error in the SLR $C_{10}$ term caused by the uneven distribution of the ground station network. Also, the difference (after inversion) was small over the combination of Greenland and Antarctica. Certainly, the geocenter term is not what is causing the divergence of SLR from GRACE after 2010, for example (I checked).

I do agree that if one was actually trying to measure the total mass loss of Greenland/Antarctica with this method, so as to compare to other similar estimates, a geocenter would be required. However, in this case, the discrepencies between SLR and GRACE are so large that the main point of the article is actually that one should NOT use 5x5 monthly SLR to push the estimate of mass change back in time. That being the case, the comparison can be run without geocenter being added (in either GRACE or SLR, to keep things equal).
* * *
*The supplement has a *.zip ending but actually is in *tgz format
* * *
Sorry; I'll fix that.
* * *
* abstract: maybe add some numbers in the abstract to quantify things a bit more
* * *
Which particular details would the reviewer care to have quantified?
* * *
* Does the average TC reader knows what is meant by 5x5, 10x10?
* * *
Good point. I'll make sure that's defined initially.
* * *
* eq 1 shouldn't the '1-' be outside of the fraction?
* * *
Corrected.
* * *
* " indicative of a systematic interannual-scale error in the SLR inversion" What is meant by this? Maybe add a reference, which illustrates the problem at hand?
* * *
What I mean is that, while the GRACE-ChengSLR trend difference is 40% the size of the total trend for 2003-2014, I do not believe this really represents an inability of SLR to represent the long-term trend. Rather, I believe this to be a symptom of SLR's tendency to veer away from the GRACE "truth" for multiple years in a row, then correct itself and come back into alignment (as it seems to be doing in 2017, and as it also may have done back in 2002). Neither I nor Minkang Cheng know of no reference which discusses this, since the accuracy of SLR's interannual variability is very hard to quantify, particularly

pre-GRACE. The 15-year record since GRACE started may not be long enough to quantify deviations which take 5+ years to resolve.

This line now reads: "So instead of representing a true, long-term error in trend, the large interannual differences between GRACE and the Cheng 5x5 SLR series are probably indicative of a systematic interannual-scale error in the SLR inversion, which cannot be well quantified given the relatively short length of the GRACE record." I hope that's clearer.
* * *
*  "451 + 28 Gt/yr" I assume this is for Antarctica and Greenland? Maybe explicitly mention this again
* * *
Yes, and done.
* * *
* Use of diagonal SLR and GRACE error-covariances , and thus neglecting off-diagonal
error-covariance. I think this is the most serious issue I can find in the paper. Since
I don't know whether this is going to have a large impact on the results I'm recommending
a major revision to allow the authors to clarify this. I suspect that in particular
SLR may have significant off-diagonal components in its error-covariances. The SLR
network is very sparse and may not be optimal for the retrieval for ice mass change
signals at higher latitudes. To account for this, one would in principle need to propagate
the full SLR error-covariance on the 1x1 degree grid used as observations. The
associated error-covariance matrix of the gridpoints will consequently be quite unstable
(e.g. from 36 SLR 'observations' one produces 360x180 observations, without adding
more information), which potentially could break down the inversion scheme as it is
implemented now. In the current setup, the authors ignored error-covariances and by
choosing an equidistant 1x1 grid also artificially increased the density of observations
at higher latitudes inversely proportional to cosine(lat). In a broad sense, ignoring offdiagonal
contributions and artificial increase of observations can be interpreted as a
regularization, which the authors should justify. I therefore, propose that the authors ei-
ther justify their choices for the 1x1 grid in combination with a diagonal error-covariance
or better: that the authors replace matrix H (see eq S1) by an operator which directly
maps Stokes coefficients to the unknown vector a. When full error-covariances are
available these can then also be implemented with hopefully relatively little effort.

* "and thus heavily dependent on the same very low degree spherical harmonics"
Maybe quantify this with formal error correlations?
* * *
I tested the impact of the Cheng 5x5 SLR covariance matrices a year or so ago, using the uncorrelated inversion method only (see example for Greenland below). The red line is the inversion with the covariance matrix included, while the blue line uses the identity matrix instead. As you can see, the subannual part changes – but that's also the least accurate part of the

signal. The long-period and interannual signals remain effectively the same. Now, this was done with an older, simpler version of the code, without the correlations between sub-regions included. But I've no reason to believe that results using the modern code would look all that different. (I should add, by the way, that one reason the error covariances make little difference is probably because the Cheng SLR series uses a consistent number of satellites in it: always just the five. So the accuracy of the combined solution doesn't drastically change when a new satellite is added/removed, as is the case in the Talpe paper.)

[Figure]

I just talked to Matt Talpe. Neither he nor I can find any way to take these spherical harmonic covariances and propagating them onto a 1x1 degree grid. The only way to take the error covariance into account would be to switch to a spherical harmonic representation from the top-down. That's simple in concept, but involves a total rewrite of the correlated inversion computer code in practice. Moreover, spherical harmonics naturally force the use of global data, while the gridded technique also allows us to use the same code on limited regions of the world. For 5x5 harmonics like SLR, that isn't useful, but it is valuable for other purposes using 60x60 GRACE data. Given the lack of clear improvement in the inverted results above, we chose to stick with the easier-to-use system already in place.

As I mentioned to the other reviewers, Minkang Cheng has very recently created an SLR series using the exact same input data, but estimating over 6 months rather a single month. When I repeat the inversion process with that data, I find that the divergence after 2010 vanishes (proving that it really was just a numerical instability). Once we tidy up these new results, we will surely write another paper. I feel it would be more meaningful to incorporate the detailed error analyses you mention, in that future paper instead, which will (hopefully) contain a timeseries which we feel is stable and trust-worthy

enough for others to use.  Given that the conclusion of this current paper is basically "the monthly SLR data is not accurate enough to use for this purpose", extended error calculations seem noncritical, to me.  I hope that a promise to look into the impact of the error covariances for the future will be sufficient for you and the editor.

5  Thank you very much for your assistance and helpful thoughts,

Jennifer Bonin

Response to Reviewer #3 (Kosuke Heki)

Thank you for your time and effort. I really appreciate the review.

5  *No. 1 Separation of Greenland and Antarctica using external information:*
   *The limited spatial resolution of the SLR 5 x 5 model could not separate ice losses from the two ice sheets. Nevertheless, I think there are external*
   *clues to answer the question, how much coming from Greenland and how much from Antarctica. Matsuo et al. (2003) used the quadratic*
   *component in the vertical position time series of GNSS stations in Greenland to validate their results. Because of uncertainties in GIA models, it is*
   *not straightforward to discuss linear uplift/subsidence rates of the Antarctic GNSS stations. However, because GIA rates do not change in a short*
10 *time-scale, quadratic (or higher degree) components in vertical position would entirely reflect the elastic response of the lithosphere to the*
   *present-day ice melting. Several GNSS station in Antarctica have been operational since 1990s, and the authors at least discuss if the signature of*
   *the accelerated ice mass loss ever exists in Antarctica.*

You, Matt King and I went back and forth a little on this via email, but to recap for the editor, while I agree that some

15 "guestimation" based on GPS is probably possible on this issue, I don't think I know enough about the subject to do it

myself. As Matt mentioned, there would be a lot of questions about mantle viscosity effects, in addition to the question of

time-variable modern-day surface loading. More, I worry that any such process would have to assume a continuation of

linearity in regions where the signal is linear now, during timespans when the signal might really have been accelerating

(etc). So this sort of separation would have to be handled very delicately, I think.

It's a good idea for someone to try to combine these data types, nonetheless, I think. For the time being, I have added a

comment in the relevant appendix section suggesting: "While it might be plausibly possible to use external sources (such as

ground GPS stations) to separate the two regions, that is likely to be a complex process, particularly as one goes backwards

in time to periods when few GPS stations exist. We leave such efforts to a future paper." Perhaps that will inspire someone.

*No 2. Reality of the departure of SLR data from GRACE:*
*Below I compare Figure 4 (left) and a figure drawn by the reviewer using the CSR Level-2 RL05 spherical harmonics data with standard filters*
*(right). It shows the gravity time series at a certain point in southern Greenland (65N 40W), and indicate anomalous changes after 2012, a short-*
*term accelerated mass loss in 2012 and a longer-term stationary behavior until present (reflecting increased precipitation there). I see*
30 *some similarity between the 5x5 SLR data (rather than GRACE HiRes-Local) and the mass changes in southern Greenland. Is it conceivable that*
*mass signals in southern Greenland leaked into the SLR 5x5 solution?*

I noticed this visual similarity, too, initially. Two things convinced me that it's not a case of "overweighting" Greenland

somehow. First, because any such signal would be seen by GRACE (as you note), but neither my "highres" case nor the 5x5

35 or global 60x60 GRACE inversions see the amount of decrease seen by SLR. (They all slow down somewhat in the later

years, please note. They just don't show the strong acceleration in 2010-2011, then plateauing that SLR does.) Secondly

and more importantly, though, as I mentioned in my email to you, last month I finally cajoled Minkang Cheng into making a

new SLR series in which he combines 6 months of data into a single solution (rather than only a month). In that SLR series,

the big deviation that his monthly solutions showed away from GRACE after 2010 totally disappears. This proves to me that

40 it was just an artifact of the SLR errors, not a real signal from Greenland or anywhere else. It seems that the monthly SLR

solutions are just barely stable, and adding more observations stabilizes them better.  So I'd like to put this question on hold for a future paper, until I can write up the results on this brand-new SLR series.  I still think it's important to put the results of the ordinary monthly solutions out there, though, since those are commonly available.  Not to mention getting the method clearly down on paper.

*Minor comments*
*Page 9 line 4: "trend errors are statistically indistinguishable from zero." sounds strange (trends could be indistinguishable from zero but errors should not be indistinguishable from zero).*
*Page 11 line 9: Please explain the "input-output method"?*
10 *Page 12 line 13: "before" what (words missing)?*
*Page 14 line 19: Nerem and Wahr (2011) missing in the reference list*

All these things have been reworded or added.

15 Thank you again for your helpful thoughts,

Jennifer Bonin

[revised manuscript text omitted]

Antarctica (Figure S2b) shows the opposite results: the simple process-noise constraint results in large trend errors, while the correlation-based constraint does a fair job of recovering the simulated input. At 5x5 resolution, the trend errors increase from 15% for the constraint-correlated technique, to 31% for the process-noise-correlated technique. The situation actually grows worse as resolution improves: for 60x60 data, the errors jump from 10% to 64%. The reason for this is straightforward and concerns the physical meaning of the computed amplitudes, **a**, over such huge areas. The different parts of the greater Greenland region tend to vary contemporaneously with each other, so that using a single number to define all of them (in a weighted fashion) will be close to correct, though not perfect. However, Antarctica does not vary so uniformly. Western Antarctica is expected to have lost mass each year recently, while eastern Antarctica is thought to be close to equilibrium and has had periods where the overall mass has likely increased in recent years (Boening et al., 2012). A single value for **a** cannot represent both situations at once. Either the two sub-regions need to separated – which works for 60x60 data but is unrealistic with only 5x5 data – or we need to limit the two halves of Antarctica in different ways. The correlation constraint technique does the latter, and results in a more reasonable estimate for the Antarctic trend because of it.

This leaves us with no technique that works well for all parts of the world. We did attempt to combine equations S3 and S4, so that we use the simple process-noise constraint over Greenland, while at the same time using the correlation constraint

over Antarctica (not shown). However, we found that the 5x5 results were still less good than using the correlation constraint everywhere. We hypothesize that the reason we cannot get both the process noise technique's good results in Greenland and the correlation constraint technique's good results in Antarctica at the same time is because the two polar areas are inseparable when looking at only the lowest degree spherical harmonics. Changing the results in one area forces

5    changes in the results of the other. While it might be plausibly possible to use external sources (such as ground GPS stations) to separate the two regions, that is likely to be a complex process, particularly as one goes backwards in time to periods when few GPS stations exist. We leave such efforts to a future paper.

Instead, given the inherent connection between Greenland and Antarctica, we choose to sum the two regions and consider

10   the combination as our final answer (Figure S2c). While our simulation demonstrates that we are unable to accurately separate Greenland from Antarctica at a 5x5 resolution, we can at least confirm that the entire polar land ice mass change can be estimated with reasonable accuracy at an SLR-like spatial resolution. Trend errors over the combined polar regions are estimated at 1.3 ± 1.6% of the trend for the correlation-constraint technique, compared to 13 ± 4% for the process-noise-constraint technique. To back out the Greenland area and Antarctic mass changes independently, we would require spherical

15   harmonics which we trust to approximately degree/order 10.

We would note that, very recently, *Talpe et al.* [2017] used a similar method to estimate Greenland and Antarctica's mass change. They used principal component analysis to determine GRACE's dominant long-term spatial pattern of variability (rather than pre-defining weighted regions as we do here), then derived the time-series associated with that pattern from

20   SLR. They only focused on the first mode, which gives the large-scale polar mass loss, since the next three modes were related to the hydrological cycle instead. This alternative technique is thus also unable to truly separate Greenland from Antarctica using SLR data. It additionally assumes stationarity between the GRACE time-frame and the pre-GRACE time-frame, such that the mass change in Greenland and Antarctica is treated as a single entity varying as one, with the relative weight between the two determined not by SLR but by the first EOF mode from GRACE's 2003-2016 data. While this is

25   likely to produce a more accurate spatial pattern than our pre-weighted regions do during the GRACE time-frame, we can not assess how well that pattern reflects the true state of polar ice mass change pre-GRACE. One benefit to the correlation-constrained sub-regions technique used here is that it allows the pre-GRACE SLR data to set the spatial pattern without influence from the melt during the GRACE years.

30   Our analysis was designed to minimize the errors of a 5x5 inversion. We found that using the same optimization parameters and regional definitions with 60x60 data resulted in an over-estimation of the mass trend of 6.5 ± 0.8%. When using real data, we do not wish this exaggerated trend to appear as part of our comparisons. Instead, when we create 'truth' series for the real-world comparison from the GRACE 60x60 data, we use the simple process-noise technique (equation S3) over each pole separately, and with higher-resolution regions to invert into. Over Greenland, we use the local regions defined in *Bonin*

*and Chambers* [2015] (excluding the specialized glacier-only basins used there), which include thirteen separate regions in Greenland itself. Over Antarctica, we use the Antarctic division shown in Figure S1f, but with the southern ocean divided into twelve sectors. By computing the inversion locally rather than globally, we are able to achieve excellent similarity to the simulated 'truth' in each region. In Greenland, the trend error is just -1.5 ± 0.3%, while in Antarctica it is 0.6 ± 0.5%.

5    Directly summing the two inverted time-series to estimate the total polar mass loss results in a trend error of only 0.7 ± 0.5%. This is proof that the inversion technique works very well with higher-resolution data. However, we are not able to accomplish the same separation with 5x5 inputs.

**Supplemental Information:  Comparison of Input Series by Harmonic Coefficient**

[Figure]

[Figure]

**Figure S3: Time-series of the GRACE (black), Cheng 5x5 SLR (red), and Sośnica 10x10 SLR (blue) input data series, by spherical harmonic coefficient. A 200-day smoother has been applied to all series.**

---

## Author Response (AR2)

I've made the desired comparison, both in the supplemental and the main text.  (I assume this is all the reviewer was looking for.)  Here are the new sentences, added after the tables in both the relevant sections:

In the main text:

This is most likely an indication of real differences in the SLR vs. GRACE data, not something caused by the processing technique itself, as trend errors from the inversion method are expected to be just $1.3 \pm 1.6\%$ (see Table S1).

In the supplemental text:

The real-world trend differences between GRACE and SLR over the combined Greenland/Antarctica area range between 7% and 41% (see Table 1 in the main text), so an expected 1.3% error caused by the technique is acceptable in comparison.

Thank you again for your time,
Jennifer Bonin